# In-House Filtration Efficiency Assessment of Vapor Hydrogen Peroxide Decontaminated Filtering Facepiece Respirators (FFRs)

**DOI:** 10.3390/ijerph18137169

**Published:** 2021-07-04

**Authors:** Khaled Al-Hadyan, Ghazi Alsbeih, Ahmad Nobah, Jeffrey Lindstrom, Sawsan Falatah, Nawarh Faran, Salem Al-Ghamdi, Belal Moftah, Rashed Alhmaid

**Affiliations:** 1Radiation Biology Section, Biomedical Physics Department, King Faisal Specialist Hospital and Research Centre (KFSH&RC), Riyadh 11211, Saudi Arabia; galsbeih@kfshrc.edu.sa; 2Emerging Technology Unit, Biomedical Physics Department, King Faisal Specialist Hospital and Research Centre (KFSH&RC), Riyadh 11211, Saudi Arabia; anobah@kfshrc.edu.sa (A.N.); bmoftah@kfshrc.edu.sa (B.M.); 3Perioperative Services Department, King Faisal Specialist Hospital and Research Centre (KFSH&RC), Riyadh 11211, Saudi Arabia; jlindstrom@kfshrc.edu.sa; 4Protocol Service Nursing Department, King Faisal Specialist Hospital and Research Centre (KFSH&RC), Riyadh 11211, Saudi Arabia; sfalatah@kfshrc.edu.sa; 5Quality Management Department, King Faisal Specialist Hospital and Research Centre (KFSH&RC), Riyadh 11211, Saudi Arabia; nfaran@kfshrc.edu.sa; 6Infection Control and Hospital Epidemiology Department, King Faisal Specialist Hospital and Research Centre (KFSH&RC), Riyadh 11211, Saudi Arabia; salghamdi6@kfshrc.edu.sa; 7General Corporate Consultancy Department, King Faisal Specialist Hospital and Research Centre (KFSH&RC), Riyadh 11211, Saudi Arabia; rhmaid@kfshrc.edu.sa

**Keywords:** COVID-19, N95 masks, KN95 masks, vapor hydrogen peroxide, FFRs, decontamination, filtration efficiency

## Abstract

To cope with the shortage of filtering facepiece respirators (FFRs) caused by the coronavirus disease (COVID-19), healthcare institutions have been forced to reuse FFRs using different decontamination methods, including vapor hydrogen peroxide (VHP). However, most healthcare institutions still struggle with evaluating the effect of VHP on filtration efficiency (FE) of the decontaminated FFRs. We developed a low-cost in-house FE assessment using a novel 3D-printed air duct. Furthermore, we assessed the FE of seven types of FFRs. Following 10 VHP cycles, we evaluated the FE of KN95 and 3M-N95 masks. The 3M-N95 and Benehal-N95 masks showed significant lower FE (80.4–91.8%) at fine particle sizes (0.3–1 µm) compared to other FFRs (FE ≥ 98.1%, *p* < 0.05). Following 10 VHP cycles, the FE of KN95 masks was almost stable (FE stability > 99.1%) for all particle sizes, while 3M-N95 masks were stable only at 2 and 5 µm (FE stability > 98.0%). Statistically, FE stability of 3M-N95 masks at 0.3, 0.5, and 0.7 µm was significantly lower (*p* ≤ 0.006) than 2 and 5 µm. The in-house FE assessment may be used as an emergency procedure to validate the decontaminated FFRs, as well as a screening option for production control of FFRs. Following VHP cycles, both masks showed high stability at 5 µm, the size of the most suspected droplets implicated in COVID-19 transmission.

## 1. Introduction

The rapid increase in novel coronavirus disease (COVID-19) patients caused a dramatic global shortage of filtering facepiece respirators (FFRs), in particular, N95 masks [1,2]. The N95 mask is a single-use respirator that can, when fitted correctly, block up to 95% of as small as 0.3 µm or larger particles. The N95 masks are approved by the US Food and Drug Administration (FDA) and by the USA National Institute for Occupational Safety and Health (NIOSH) [3,4]. The FFRs, including N95 masks, are considered a critical component of infection prevention and control by decreasing COVID-19 transmission from patients to health care professionals. However, N95 masks are designed for single usage, and in cases of infectious agents, it may also be used for a single patient [5,6]. Therefore, under the massive demand for FFRs during the COVID-19 pandemic, NIOSH issued a recommended guidance for the extended use and limited reuse of N95 respirators in an emergency health care setting [6]. Although the NIOSH guidance could significantly reduce the consumption of N95 masks during the pandemic, two studies were concerned with safety of the extended use of N95 FFRs [7,8]. These concerns were related to the long survival of the SARS-CoV-2 virus on the outer layer of N95 masks, resulting in cross-contamination through direct contact between staff or indirectly by touching contaminated surfaces [7,8]. Therefore, these decontamination methods may be needed for future pandemics.

There are more than five well-characterized N95 decontamination procedures, including vapor hydrogen peroxide (VHP), moist heat incubation, microwave oven, ultraviolet germicidal irradiation (UVGI), gamma radiation, and ethylene oxide [9,10,11,12,13]. Although the USA Centers for Disease Control (CDC) has not approved the routine decontamination of N95 masks, it released emergency guidelines on the N95 decontamination methods, indicating that UVGI, moist heat, and VHP have shown the most promising results [10].

There are advantages and disadvantages to each of the three previous decontamination procedures [14,15,16]. The UVGI and moist heat procedures are ubiquitous, inexpensive, accessible, with high throughput and ease of use and no chemical residues left after the decontamination procedures [15]. However, concerns were raised by a number of researchers regarding the ability of UVGI to sterilize throughout the thickness of the FFRs, and thermal deformation and shadowing was observed on the FFRs following UVGI treatment [15,17]. The moist heat decontamination procedure, either by using moist heat incubation or autoclave, seemed to cause adverse effects on filtration efficiency (FE) and airflow of FFRs [16]. Several studies showed that the VHP decontamination procedure had no adverse effect on either FE, airflow, physical characteristic, or fit test of FFRs, with high germicidal efficacy [14,18]. Therefore, the VHP decontaminant procedure was the most interesting procedure to be locally applied among other decontaminant procedures.

In response to the shortage of FFRs, the FDA has issued three emergency use authorizations (EUA) for the emergency use of VHP to decontaminate FFRs using STERIS, Stryker and STERRAD systems [19,20,21]. The STERIS sterilization systems can be applied up to 10 times for each N95 mask using one of five models of STERIS sterilizers (V-PRO 1 Plus, maX, maX2, 60, and V-PRO s2) [22]. By using the NIOSH test procedure to measure FE [23], the 3M company showed that the VHP-STERIS decontamination method does not affect the FE in seven models of 3M-N95 masks; 1860, 1860S, 1870+, 8110S, 8210, 9205+, and 9210+ [24].

The STERIS sterilization system was selected among the other two systems to be applied in this study for two reasons. The first was the availability of four STERIS sterilizer units (STERIS V-PRO maX model) compared to a single STERRAD sterilizer unit (100NX model) at the perioperative services department in our institution, while the Stryker system was unavailable. The second reason was the limited number of approved-VHP cycles (two cycles maximum) obtained using the STERRAD sterilization system compared to the STERIS sterilization systems (10 cycles maximum) [19,21].

Although the VHP-STERIS decontamination is an FDA–EUA-approved method, in-house FE validation of the decontaminated N95 masks is needed as a quality assurance process. Currently, most healthcare institutions are struggling with how to validate the FE of decontaminated N95, as the NIOSH test procedure is not widely available, in addition to its high cost, time-consumption, and complexity [25]. Therefore, a number of healthcare institutions have developed in-house methods to evaluate the FE of decontaminated masks using the available equipment and techniques as alternatives to the NIOSH method [9,26,27].

In May 2020, during the peak of COVID-19 pandemic in Saudi Arabia, we successfully implemented the STERIS sterilization systems to decontaminate N95 masks as a crisis capacity strategy. The current study aimed to develop a simple, quick, and low-cost in-house FE assessment procedure that can be used as a screening option for production control of FFRs andas a quality assurance process, of decontaminated FFRs. Furthermore, the study aimed to assess the FE of a number of FFRs, some of which were N95 masks, and evaluate the effect of VHP-STERIS decontamination method on FE of N95 and KN95 masks.

## 2. Materials and Methods

### 2.1. Filtering Facepiece Respirators (FFRs)

The FFRs used in this study are described in Table 1. Briefly, seven types of FFR masks were used. A set of five each of Gerson 1730, Medline, Benehal, N99/N95 SpectraShield Plus, and KN95 (duck shape) masks were included. In addition, a set of 12 each of 3M-N95 (3M-8210) and KN95 (molded shape) masks were also tested.

### 2.2. Filtration Efficiency (FE) Measurement 

The purpose of using 3D-printing technology was to create a specialized air duct that could be used to assess FE of different mask types with no air leak during the FE assessment. The 3D-printing technology was employed to produce each part of the air duct with a precise shape to serve specific functions. Although such designs can be manufactured in mechanical engineering facilities, 3D-printing was the most accessible and available option during the COVID-19 lockdown.

As shown in Figure 1, a novel specialized air duct was designed using 3D-printing technology to measure the FE of different mask types.

The 3D digital design, in STereoLithography (STL) format, of the air duct is available in Appendix A. Briefly, the air duct was made by Prusa 3D-printer (model MK3S, product ID: PRI-MK3S-COM-ORG-PEI) using Polylactic Acid (PLA) filaments (Prusa Research, Prague, Czech Republic, product ID: FLM-PLA-175-YEL) and designed by two softwares; Shapr3D modelling software (https://www.shapr3d.com/ accessed on 3 July 2021) and Slic3R slicing software (https://slic3r.org/ accessed on 3 July 2021), used to generate the G code required to print the model by the Prusa printer with a dimension of 19-cm-long, 14-cm-wide, and 12-cm-high. The air duct consisted of two parts, head and tail, that could be tightly joined together by three mold bolts to squeeze a mask in a sandwich manner. Several air duct designs were fabricated and tested to ensure no air leak was present between both parts. The source of the measured particles was the ambient particulate matter (PM) air. An electrical fan flowed the PM air through the air duct tail. The fan was controlled by a variable transformer (model: 3PF1010, Staco Energy Production Company, Miamisburg, OH, USA) fixed at 47 volts to give a face velocity of 0.4 m/s; measured by Velocicalc Air Velocity Meter 9545 (TSI, Shoreview, MN, USA, product ID# 9545-A). The head of the air duct was designed to fully contain a stainless steel part of an AeroTrak particle counter (TSI, Model 9306, product ID: SKU 9306-03). The AeroTrak particle counter is an optical-based particle counter that measures the particle number concentration at six different particle sizes (0.3, 0.5, 0.7, 1, 2, and 5 µm) using light scattering, a method that is based upon the amount of light deflected by a particle passing through the detection area of the counter [28]. The AeroTrak particle counter was calibrated for each use by Purge (Zero Count) Filter (TSI, product ID# 700005). The air particles entered the air duct from the tail to the head and penetrated the squeezed mask at a flow rate of 2.8 L/min. The particle number concentration of the PM air was assessed at least five times before FFR assessment. For each particle size, FE of each mask was assessed for 1 min and calculated using the following formula:$$\mathrm{FE}\;\left(\%\right)=100-\left(\frac{\mathrm{number}\;\mathrm{of}\;\mathrm{penetrated}\;\mathrm{particles}\;}{\mathrm{average}\;\mathrm{number}\;\mathrm{of}\;\mathrm{particles}\;\mathrm{in}\;\mathrm{air}}\times 100\right)$$

Although the NIOSH’s standard sampling time is 10 min, the 1-min sampling time was selected based on an early pilot experiment performed in our laboratory. The experiment showed no overall significant difference between FE sampling times of 1 and 10 min of either KN95 (n = 5) or N95-3M (n = 5) FFRs with *p*-values of 0.757 and 0.699, respectively (Appendix A). Therefore, the 1-min sampling time was selected since it is more suitable for such an in-house method, especially with a large number of FFRs and VHP cycles, as shown later.

### 2.3. Filtration Efficiency (FE) Following VHP Cycles

For FE evaluation, 12 masks of each 3M-N95 and KN95 were subjected to 10 VHP cycles using STERIS sterilization systems (Mentor, OH, USA). The full protocol of VHP-STERIS sterilization systems can be found on the STERIS website [29]. Briefly, at the FFRs collection station, the used FFRs were carefully labelled and pouched in a VHP sterilization pouch (Tyvek, Product ID#12340). Before sealing the pouch, a VHP indicator strip (Sterrad, Irvine, CA, USA, product ID#14100) was inserted in the pouch to confirm the masks’ VHP exposure. A maximum of 10 pouches could be processed per each non-lumen cycle (28 min for each cycle) using the STERIS V-PRO maX sterilizers (STERIS).

The FE assessment of decontaminated FFRs was conducted 10 times on 10 different days for 10 VHP cycles; therefore, the PM air particle number concentration was different for each day of the experiment. To minimize these differences, we conducted the 10 FE assessments at the same place using the same particle counter. Overall, these minor differences were relatively small and statistically considered. After each VHP cycle, the stability of FE for each particle size was calculated using the following formula:$$Stability\;of\;FE\;\left(\%\right)=\left(\frac{FE\;at\;cycle\;X\;}{FE\;of\;control\;\left(cycle\;0\right)}\times 100\right)$$

### 2.4. Statistical Analysis

To examine statistical differences in FE and FE stability of FFRs for different particle sizes and over multiple VHP decontamination cycles, the parametric one-way pepeated measures analysis of variance test was used to detect significant differences in the means values for each set of data. In case the data did not pass the Shapiro–Wilk normality test, the nonparametric Friedman repeated measures analysis of variance on ranks test, which compares medians, was applied. The percent uncertainty of the particle counter at each particle size was calculated by dividing the absolute uncertainty (standard error) by the average of measurements multiplying by 100. All the statistical analyses were performed using SigmaPlot 12.5 for Windows (SPSS Inc., Chicago, IL, USA).

## 3. Results

### 3.1. Filtration Efficiency (FE) of FFRs

As shown in Figure 2, KN95 (molded shape), Gerson 1730, Medline N95, N99/N95 SpectraShield, and KN95 (duck shape) masks showed an FE of 98.1% or above at all particle sizes. At 2 and 5 µm particle diameters, all FFRs, except Benehal-N95, showed an FE of 95% or above. The 3M-N95 and Benehal-N95 masks were less efficient (FE = 80.4–91.8%), particularly at low particle diameters (0.3–1 µm), than other FFRs (FE = 98.6–99.9%). Statistically, the overall FE values of FFRs failed to pass the normality test, therefore, the median FE values were used for statistical analysis. The Friedman repeated measures analysis of variance on ranks showed that the overall median FE values for the 3M-N95 and Benehal-N95 masks were significantly different from the N99/N95 SpectraShield mask (*p* < 0.05, respectively) with no other significant differences observed.

### 3.2. Filtration Efficiency (FE) Stability of N95 and KN95 Masks Following VHP-STERIS Sterilization Systems

The FE stability of the two FFRs (KN95 and 3M-N95) masks was studied for up to 10 VHP-STERIS sterilization cycles (Figure 3). Results showed that for all sizes of particle diameters, the FE of KN95 masks was almost stable (FE stability > 99.1%) over 10 VHP cycles. However, FE of 3M-N95 masks was stable only at particle diameters of 2 and 5 µm (FE stability > 98.0%), while it showed wide variation in FE stability (73.9–97.0%) at particle diameters between 0.3 and 1 µm. Statistically, the 3M-N95 masks showed an overall significant (*p* < 0.001) reduction in FE stability with increased VHP cycles using the Friedman repeated measures analysis of variance on ranks test. In particular, FE stability of 3M-N95 masks at 0.3, 0.5, and 0.7 µm over 10 VHP cycles was significantly lower than FE stability at 5 µm with *p* values of <0.001, <0.001, and 0.006, respectively. Similarly, FE stability of 3M-N95 masks at 0.3 and 0.5 µm over 10 VHP cycles was significantly lower than FE stability at 2 µm with a *p*-values of 0.002 for both.

## 4. Discussion

The procedure of decontamination and subsequent reuse of N95 masks is recommended to be applied only as a crisis capacity strategy with the objective of making healthcare institutions self-sufficient for N95 practice [6]. This study provided a simple, quick, and low-cost in-house method for the FE assessment of FFRs. Furthermore, the study evaluated the FE of different FFR types pre- and post-VHP decontamination procedure.

The parameter comparison between the in-house FE assessment method and NIOSH FE assessment method is presented in Table 2 [23].

The in-house FE assessment was developed as an emergency alternative to NIOSH FE method to reduce the SARS-Cov-2 spread. In the in-house FE assessment, the FFRs were exposed to the (PM) air rather than artificial aerosol clouds. Furthermore, the in-house FE assessment does not assess very small particles (<0.3 µm), which are way below the sizes (2–5 µm) suspected of transmitting viruses via expelled droplets during speaking and coughing [30,31]. Although the airflow rate of the in-house FE assessment (2.8 L/min) is lower than NIOSH FE method (85 L/min), it is more compatible with the exhaled-breath flow rate for healthy people (5–10 L/min) than the NIOSH method [32]. The AeroTrack particle counter used in this study (Figure 1) has been used in several studies to assess FE of FFRs as a low-cost alternative to the TSI automated filter tester [9,26,27]. 

The cost of the in-house FE assessment is insignificant compared to the approved assessments for two reasons. The first reason is the low cost of the recent handheld particle counters such as the AeroTrak (TSI 9306) particle counter or its equivalent compared to the stationary particle counter. Briefly, the high cost of the stationary particle counter has been a barrier to long-term PM air monitoring and studies [33]. However, a recent generation of low-cost handheld particle counters showed high monitoring of ambient air particles in indoor and outdoor environments with high spatiotemporal resolution [34]. The second reason is the availability of handheld particle counters in most hospitals as they are widely used to assess indoor air quality either in the operation theatre or patients’ rooms [35,36]. Therefore, the handled particle counters can be employed for FE assessment of decontaminated FFRS in a pandemic with no extra cost. In terms of 3D-printing cost, the total approximate cost of the filaments used to fabricate the 3D-printed air duct is less than $10, as the 3D-printer (~$750) and the slicing software ($20) are available.

Results showed that five out of seven FFRs retain an FE capacity of >95% at particle sizes of 0.3 µm or larger (Figure 2). Only two types of FFRs (3M-N95 and Benehal) displayed statistically significant (*p* < 0.05) reduced FE (80.4–91.8%), particularly at fine particle sizes (0.3–1 µm), compared to other FFRs (FE = 98.6–99.9). In agreement with our findings, a recent study showed that 3M-N95 masks displayed an FE of 88.1, 91.4, and 89.2% at particle sizes of 0.3, 0.5, and 1 µm, respectively [27]. Another study also showed that a similar type of 3M-N95 mask (3M-1801) has an FE of ~85% at 0.3 µm [37].

The size of the SARS-CoV-2 virus is between 0.07 µm to 0.09 µm; however, it is transmitted through respiratory droplets (>5 µm) that are released from infected patients during coughing, sneezing, or speaking and then transmitted to another healthy person by fomite transmission [38,39,40]. Although airborne SARS-CoV-2 transmission, caused by the transmission of droplet nuclei particles (<5 µm), was not implicated by the World Health Organization to be the main transmission route of SARS-CoV-2, recent reports indicated that droplet nuclei particles might play a more prominent role in COVID-19 infection, in particular, during medical procedures that generate aerosols [41,42,43,44]. A recent report using data from a number of experimental studies and theoretical models showed that the minimum size of droplets that can contain and then transmit SARS-CoV-2 was calculated to be approximately 4.7 µm [31]. Therefore, the current findings may not, hypothetically, affect the ability of the two FFRs (3M-N95 and Benehal) to prevent SARS-CoV-2 transmission; further research is needed.

While the FE of KN95 mask was stable over 10 VHP cycles, the 3M-N95 was only stable at the highest (2 and 5 µm) particle diameters (Figure 3). The 3M-N95 appeared somewhat unstable for smaller particles (<2 µm) with increasing VHP cycles. A recent study showed that the FE of KN95 and another model of 3M mask (3M-1860) were stable after a single VHP cycle at a particle size of 1 µm, which is compatible with our findings [45]. Another study also observed a stable FE (only 0.4% reduction in FE) of 3M-1860 N95 following a single STERIS-VHP cycle using the NIOSH assessment method to evaluate FE [46]. Furthermore, four reports confirmed the latter studies and showed that 3 to 10 VHP cycles did not affect the FE stability of different types of 3M-N95 masks [47,48,49,50].

### 4.1. Uncertainties and Repeatability of the Experimental Measurements

The percent uncertainty of the particle counter (AeroTrak TSI 9306), given at one standard deviation, was calculated based on the standard error for 10 measurements of the ambient air assessed for each batch of particle size. These 10 measurements were obtained from the control measurements of the third VHP cycle, and they were selected due to the high number of control measurements assessed at the same time on the same day. As shown in Table 3, the percent uncertainties of our particle counter were acceptable and comparable to the reference uncertainties of the manufacturer calibration at all particle sizes [28,51]. The uncertainties related to other physical parameters, such as pressure, flow rate, and measurement time, could be assumed as negligible as they were included in the particle counter parameters and considered in the overall uncertainties.

The standard error was used to evaluate the repeatability of the experimental measurements in our study. In Figure 2, the highest standard error was 3.6, observed for the average FE assessment (87.1%) of Benhal N95 masks at a particle size of 0.3 µm (Appendix A). In Figure 3 (left panel), the highest standard error was 0.4, observed for the average FE assessment (99.6%) of KN95 masks at a particle size of 0.7 µm, while in the right panel, the highest standard error was 4.9, observed for the average FE assessment (87.1%) of N95-1820 masks at a particle size of 0.5 µm (Appendix A).

### 4.2. Study Limitations

The study has limitations. Only two types of FFRs and one VHP sterilization system have been tested. The in-house FE assessment has less sensitive parameters than the NIOSH FE method regarding airflow, pressure, and sampling time. Furthermore, the NIOSH method used artificial aerosol clouds to assess FE, which seem more sensitive than ambient particulate matter air.

## 5. Conclusions

Although the in-house FE assessment seems less restrictive than the NIOSH FE method, it may have a potential benefit as an emergency procedure to validate the FE of decontaminated FFRs where the NIOSH method is unavailable. Furthermore, the in-house FE assessment can be used as a screening option for production control of FFRs as it is faster and cheaper than the standardized procedures.The KN95 mask showed higher FE stability than 3M-N95 masks at small particle sizes (0.3–2 µm) following VHP cycles. However, both masks showed high FE stability at 5 µm, the size of the most suspected droplets implicated in COVID-19 transmission; however, further research is needed. For future research, monitoring the potential changes in the physical properties of decontaminated FFRs, such as size, color, solidity, and smell, should be considered [52]. Furthermore, the electrostatic charge analysis in the filtration layer within FFRs following VHP cycles should be considered a new aspect of FFR evaluation in further research [53]. Additionally, the 3D-printed specialized air duct could be used to evaluate other FFR decontamination methods and the efficiency of non-medical masks such as surgical and cloth masks.

## Figures and Tables

**Figure 1 ijerph-18-07169-f001:**
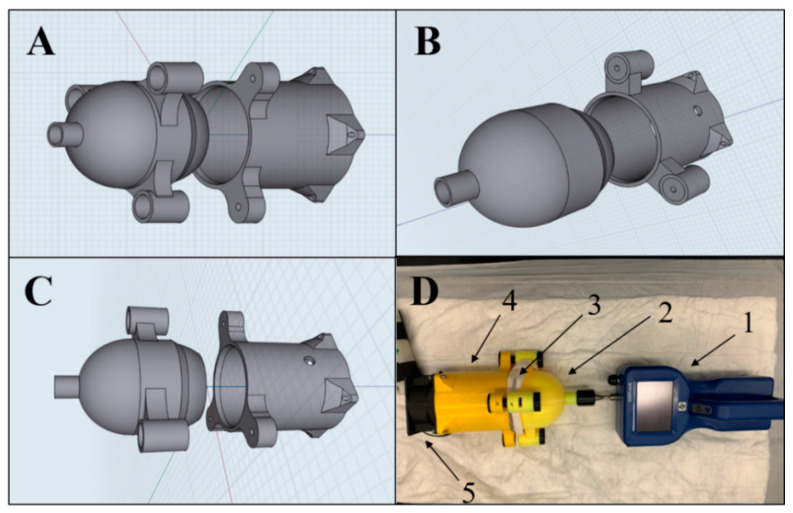
Illustration of the 3D-printed specialized air duct used to measure the filtration efficiency (FE) of FFRs. (**A**–**C**) are 3D designs from different angles of the air duct. (**D**) is the actual air duct connected to AeroTrak particle counter (1), through the head (2) of the air duct showing a tested mask (3), bound to the tail (4) of the air duct and the electric fan (5). The overall dimensions of the air ducts are 19-cm-long, 14-cm-wide, and 12-cm-high; the head dimensions are 10-cm-long, 14-cm-wide, and 12-cm, while the tail dimensions are 9-cm-long, 14-cm-wide, and 12-cm.

**Figure 2 ijerph-18-07169-f002:**
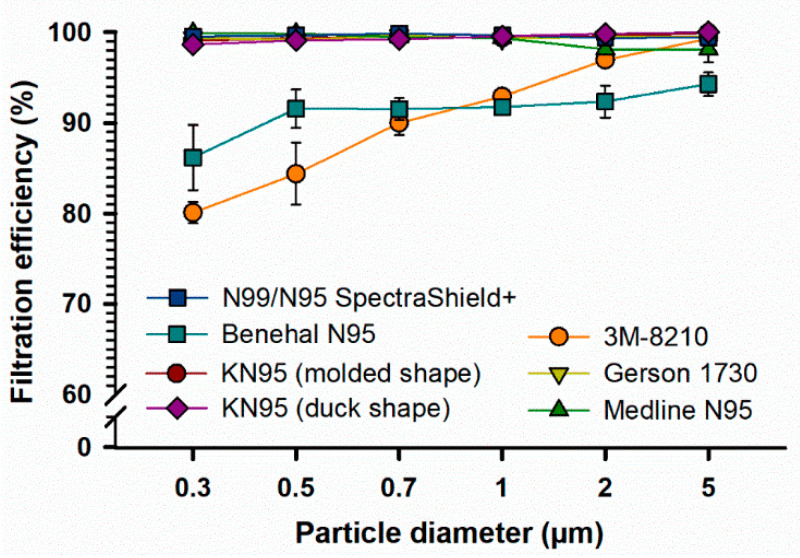
Filtration efficiency (FE) of different types of FFRs. Seven types of FFRs were applied to FE evaluation for different particle sizes (0.3–5 µm) using AeroTrak particle counter; KN95 (moulded shape, n = 10), 3M-1820 (n = 10), Gerson-1730 (n = 5), Medline (n = 5), Benehal (n = 5), N99/N95 SpectraShield Plus (n = 5), and KN95 (duck shape, n = 5) masks. Symbols represent the mean, and error bars indicate the standard error. Statistically, an overall significant difference (*p* = 0.001) was observed between the median FE values between the FFRs, with a *p*-value of 0.001. Pairwise comparison: 3M-N95 vs. N99/N95 SpectraShield: *p* < 0.05; Benehal-N95 vs. N99/N95 SpectraShield: *p* < 0.05; no other pairwise comparison significant differences were observed. Raw data are listed in Appendix A.

**Figure 3 ijerph-18-07169-f003:**
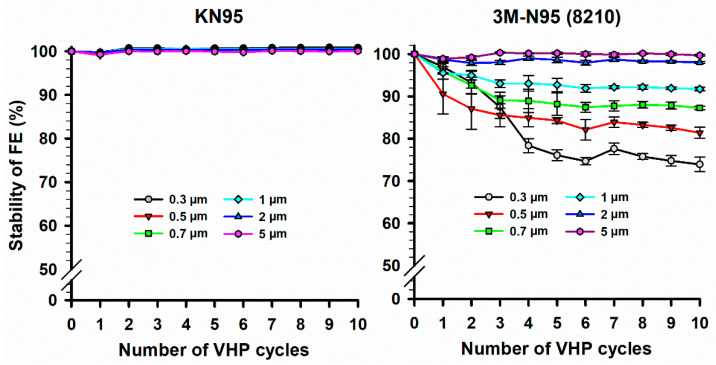
Stability of filtration efficiency (FE) of KN95 and 3M-N95 masks over H2O2 decontamination cycles. Twelve of each KN95 (left panel) and 3M-N95 (right panel) masks were cycled in VHP-STERIS sterilization systems 10 times. The masks were applied to FE stability evaluation after each VHP cycle for different particle sizes (0.3–5 µm) using AeroTrak particle counter. Symbols represent the mean, and error bars indicate the standard error. Statistically, the 3M-N95 masks showed an overall significant (*p* < 0.001) reduction in FE stability with increased VHP cycles. Pairwise comparison: 0.3 µm vs. 5 µm, *p* < 0.001; 0.5 µm vs. 5 µm, *p* < 0.001; 0.7 µm vs. 5 µm, *p* = 0.006; 0.3 µm vs. 2 µm, *p* = 0.002; 0.5 µm vs. 2 µm, *p* = 0.002; no other pairwise comparison significant differences were observed. Raw data are listed in Appendix A.

**Table 1 ijerph-18-07169-t001:** Description of FFRs used in the study.

FFRs	Type	Valve Yes/No	Company	Country	Lot Number	NIOSH Approved
Gerson 1730 Particulate Respirator	N95	No	Louis M. Gerson	United States	TC-84A-0160	Yes
Medline-Cone Style	N95	No	Medline	United States	TC-84A-5411	Yes
Benehal Particulate respirator face mask	N95	Yes	Suzhou Sanical Protective Product	China	541529	Yes
SpectraShield+	N99/N95	No	Nexera Medical	United States	A84740.B1	Yes
KN95 (duck shape)	KN95	No	Yuyao Yukang Medical Equipment	China	20200506	No ^1^
3M-8210 ^2^	N95	No	3M	United States ^3^	A12199	Yes
KN 95 (molded shape)	KN95	No	ZhongShan XiaoLan YiShuai Gament Factory	China	2020042701	No ^1^

^1^ Meets Chinese standards GB2626:2006; ^2^ FDA-EUA-approved for decontamination and reuse; ^3^ Multinational conglomerate corporation.

**Table 2 ijerph-18-07169-t002:** Comparison between in-house and NIOSH filtration assessment methods.

	In-House Method	NIOSH Method
Particle size (µm)	≥0.3 *	≥0.075
Flow rate (L/min)	2.8	85.0
Aerosol concentration (mg/m^3^)	Ambient particulate matter air (20–50)	200
Testing filter	TSI AeroTrak particle counter	TSI 8310 automated filter tester
Sampling time/FFR (minute)	1	10

* 0.3–5 µm.

**Table 3 ijerph-18-07169-t003:** Comparison between percent uncertainty between the current study and the manufacturer calibration of AeroTrak (TSI 9306) particle counter.

Particle Size (µm)	Percent Uncertainty (%)	Manufacturer’s Percent Uncertainty (%)
0.3	1.4	3.9
0.5	3.0	3.9
0.7	4.7	3.9
1	3.1	3.9
2	4.5	3.8
5	4.4	3.8

## Data Availability

The raw data supporting the conclusions of this study will be made available by the authors upon request, without undue reservation.

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
