# Peer review of "In-House Filtration Efficiency Assessment of Vapor Hydrogen Peroxide Decontaminated Filtering Facepiece Respirators (FFRs)"

_ijerph, 2021, doi:10.3390/ijerph18137169_

Round 1
Reviewer 1 Report
There are main concern n the paper entitled “ In House Filtration Efficiency Assessment of VHP Decontami-2 nated Filtering Facepiece Respirators (FFRs) “ that hesitate me to accept this paper
In the title, write the full name of the VHP
- There is no point in using or not using the 3D printing technology
- The particle characterization is not clear, how the particle has been characterized
- The source of the particles is not stable or characterized
- There is no control on the assessment
Reviewer 2 Report
The following deficiencies should be addressed before considering this work for publication.
- Many acronyms have been used in the paper, making the text hard to follow. It is recommended to provide a nomenclature in the revised version.
- Repeatability of experimental measurements should be discussed in the paper.
- Uncertainties associated with the experimental measurements should be analysed, reported and discussed in the paper.
- Product IDs are used in the paper. Specifications of those products should be provided in the paper for later usage.
- Dimensions should be reported for parts shown in figure 1.
- Details of the statistical analysis should be reported in the paper to ensure the reproducibility of the present results.
- It is recommended to scale the vertical axes reasonably to clearly show the values shown in the plots.
- It is highly recommended to provide the raw data used in figure 2 and 3 in the supplementary materials for future use.
- The authors should also comment on the costs involved to produce their proposed filtering facepiece respirators.
Reviewer 3 Report
This research aims at 1) developing a simple, quick, and low-cost in-house FE assessment procedure, as a quality assurance process, of decontaminated FFRs; and 2) assessing the FE of a number of FFRs, some of which are N95 masks, and evaluate the effect of VHP-STERIS decontamination method on FE of N95 and KN95 masks — being the results for the first objective considered as a significant contribution to the research in the area, while the second a minor, as it just confirms other researchers' findings.
We believe that the alternatives for NIOSH's method are not only interesting for low-income level users/countries, but they are also fundamental for developing new ideas on how the method can be improved in further revisions.
- Introduction
COMMENTS
Line 64, the authors should add one paragraph comparing advantages and disadvantages for UVGI, VHP, and moist heat, before discussing the FDA's emergency use authorization (EUA). This paragraph might prove useful for supporting the authors' argument on choosing to evaluate VHP. Please, see below some references that can help to define these points in favor and against each method:
UVGI – 10.3390/polym13050801
VHP – 10.1016/j.jhin.2020.08.005
Moist heat – 10.1016/j.jhin.2020.08.016
After that, still before discussing the STERIS method, the authors could at least cite to readers the existence of other EUA using VHP, such as the Stryker Instrument's Sterizone VP4 Sterilizer's EUA (https://www.fda.gov/media/136976/download), and the Advanced Sterilization Products, Inc. (ASP) STERRAD 100S, NX, and 100NX Sterilization Systems' EUA (https://www.fda.gov/media/136884/download).
Before line 73, where the authors start discussing their in-house FE-validation, authors should, in a new paragraph, critically evaluated their choice for STERIS instead of other systems. This evaluation might give additional support to their selection.
- Materials and Methods
Minor changes
Line 101, Figure 1, one should read "angles" instead of "angels."
Line 137, the authors mention, "In case the data did not pass the normality test." The authors should provide the name of the test, as there are many in the literature.
Either bold or not the formulae.
COMMENTS
In line 111, the authors mention, "Several optimizations were conducted to ensure no air leak between both parts." These optimizations should be explained either in the main text body, after this statement, or a specific section in the Supplementary Material.
- Results
Minor changes
Lines 162-3, the authors mention, "Statistically, the 3M-N95 masks showed an overall significant (P<0.001) reduction in FE stability with increased VHP cycles." However, they do not indicate which test they used (One-way ANOVA or Friedman's). For the sake of clarification, this information must be clear.
- Discussion
Minor changes
Line 181, one should read "parameter" instead of "parametric."
Line184, Table 2, one should read "Comparison" instead of "comparison."
Table 2, unities should be placed after each Parameter, like in "Flow rate" or "Testing time," and excluded from the following columns.
Lines 205 and 207, one should read "μm" instead of "um."
The authors indicate, "Although droplet nuclei particles (<5 um) were not implicated by World Health Organization (WHO) to cause "airborne SARS-CoV-2 transmission" [30]", but here (https://www.who.int/news-room/commentaries/detail/transmission-of-sars-cov-2-implications-for-infection-prevention-precautions), WHO considers the possibility of aerosol transmission.
COMMENT
It is not clear why the authors opted to test each FFR in their in-house method for only one minute instead of 10, just like NIOSH's method. This parameter selection should be discussed in Material and Methods, and this difference in testing time should also be considered for creating the dispersion within results.
- Conclusions
COMMENTS
There are some avenues for future research, as chemical decontamination methods can leave in those FFRs allergenic-residues and odor, which could impede the reuse of healthcare workers. It might be possible that such a decontamination method had deformed the FFRs; thus, we would suggest observing these parameters in future research. Additionally, this 3D printed specialized air duct manufactured by the authors could be used in evaluating other sterilization methods. Ergo we also suggest this as a future avenue for research.
Additionally, authors could indicate potential limitations in their paper, for instance, considering both methods (in-house and NIOSH's), as it seems that NIOSH's is more forceful to the FFRs than theirs considering the parameters. We discussed this point because one of the objectives was "to develop a simple, quick and low-cost in-house FE assessment procedure, as a quality assurance process, of decontaminated FFRs."
Departing from the idea that NIOSH's may be a more restrictive test than the in-house, we are not confident on this assertive "Therefore, both 3M-N95 and KN95 are expected to protect against infection over 10 cycle of VHP-decontamination." unless other results in the literature support it. Please, see below some references:
10.1016/j.ajic.2020.06.194
10.1371/journal.pone.0243965
10.1016/j.jhin.2020.10.007
10.1016/j.jhin.2020.12.006
References
Minor changes
The authors should use the peer-reviewed versions of the following papers:
Chin, Alex, Julie Chu, Mahen Perera, Kenrie Hui, Hui-Ling Yen, Michael Chan, Malik Peiris, and Leo Poon. "Stability of Sars-Cov-2 in Different Environmental Conditions." medRxiv (2020). 10.1016/S2666-5247(20)30003-3
Cramer, Avilash, Enze Tian, H Yu Sherryl, Mitchell Galanek, Edward Lamere, Ju Li, Rajiv Gupta, and Michael P Short. "Disposable N95 Masks Pass Qualitative Fit-Test but Have Decreased Filtration Efficiency after Cobalt-60 Gamma Irradiation." medRxiv (2020). 10.1001/jamanetworkopen.2020.9961
Cramer, Avilash K, Deborah Plana, Helen Yang, Mary M Carmack, Enze Tian, Michael S Sinha, David Krikorian, David Turner, Jinhan Mo, and Ju Li. "Analysis of Steramist Ionized Hydrogen Peroxide Technology in the Sterilization of N95 Respirators and Other Ppe: A Quality Improvement Study." medRxiv (2020). 10.1038/s41598-021-81365-7
After the peer-review process, do selected results sustain? If yes, the authors should only change the references to the peer-reviewed article. If not, change the text accordingly.
There is a current version of this article:
Oral, Ebru, Keith K Wannomae, Rachel L Connolly, Joseph A Gardecki, Hui Min Leung, Orhun K Muratoglu, Anthony Griffiths, Anna N Honko, Laura E Avena, and Lindsay GA McKay. "Vapor H2o2 Sterilization as a Decontamination Method for the Reuse of N95 Respirators in the Covid-19 Emergency." medRxiv (2020).
Does this new version change the results used? If yes, change the text accordingly. If not, provide the most recent version in the reference section.
